# Recent Advances in Flexible Piezoresistive Arrays: Materials, Design, and Applications

**DOI:** 10.3390/polym15122699

**Published:** 2023-06-16

**Authors:** Shuoyan Xu, Zigan Xu, Ding Li, Tianrui Cui, Xin Li, Yi Yang, Houfang Liu, Tianling Ren

**Affiliations:** 1School of Integrated Circuit, Tsinghua University, Beijing 100084, China; 2Beijing National Research Center for Information Science and Technology (BNRist), Tsinghua University, Beijing 100084, China; 3Center for Flexible Electronics Technology, Tsinghua University, Beijing 100084, China

**Keywords:** pressure sensor, piezoresistive sensor, flexible array, human-interactive system, healthcare

## Abstract

Spatial distribution perception has become an important trend for flexible pressure sensors, which endows wearable health devices, bionic robots, and human–machine interactive interfaces (HMI) with more precise tactile perception capabilities. Flexible pressure sensor arrays can monitor and extract abundant health information to assist in medical detection and diagnosis. Bionic robots and HMI with higher tactile perception abilities will maximize the freedom of human hands. Flexible arrays based on piezoresistive mechanisms have been extensively researched due to the high performance of pressure-sensing properties and simple readout principles. This review summarizes multiple considerations in the design of flexible piezoresistive arrays and recent advances in their development. First, frequently used piezoresistive materials and microstructures are introduced in which various strategies to improve sensor performance are presented. Second, pressure sensor arrays with spatial distribution perception capability are discussed emphatically. Crosstalk is a particular concern for sensor arrays, where mechanical and electrical sources of crosstalk issues and the corresponding solutions are highlighted. Third, several processing methods are also introduced, classified as printing, field-assisted and laser-assisted fabrication. Next, the representative application works of flexible piezoresistive arrays are provided, including human-interactive systems, healthcare devices, and some other scenarios. Finally, outlooks on the development of piezoresistive arrays are given.

## 1. Introduction

Pressure sensing is indispensable in a wide range of scenarios, e.g., physiological signal detection, environmental signal monitoring, etc. The rigid single-point sensor, such as load cells, have been maturely applied with high accuracy and stability for decades, integrated with dedicated designs. However, their large volume limits their spatial arrangement and installation. With the development of micro/nanomaterials and design strategies, endowing pressure sensors with flexibility and high spatial resolution properties have been hot research topics over the years, which not only are essentials in frontier scientific and technological fields (e.g., robotic tactile perception and prosthesis [1,2], human-interactive interfaces [3,4], wearable healthcare devices [5,6], etc.) but also have the potential as an alternative and complement to traditional single-point sensors. Flexibility enables easy and fine attachment on curved surfaces and less discomfort for biological measurement. The arrayed structure provides spatial pressure distribution, thus being able to extract abundant information. There are multiple mechanisms for flexible pressure sensing arrays that convert pressure into different electrical quantities, including piezoresistive [7,8], capacitive [9,10], piezoelectric [11,12], triboelectric [13,14], etc. Among them, piezoelectric and triboelectric sensors demonstrate high sensitivity to dynamic pressure; however, they have more complicated electronic designs and suffer from inaccuracy when measuring static pressure. For measurements including both static and dynamic pressure, capacitive sensors and piezoresistive sensors are mostly preferred. Although capacitive sensors can achieve high sensitivity and stability, they are susceptible to parasitic capacitance and noise. In comparison, piezoresistive sensors have the advantages of easy material preparation, simple readout circuits, low-cost, and anti-electrical interference capacity in measurements. 

In recent years, many studies on flexible piezoresistive sensors have been published, with new materials and structures developed to improve sensor performance. For the design and manufacture of flexible arrays, it is not just a matter of spatially arranging measuring points. It is necessary to comprehensively consider the problems that may be encountered, especially the crosstalk problem after arraying, which will lead to the mutual influence of signals between measuring points, causing the actual spatial resolution to be lower than the spacing of the array elements. Furthermore, many manufacturing methods in laboratory studies are based on manual methods, which limits their mass production, so more automation and mature equipment processes need to be introduced to promote the application of piezoresistive arrays. 

Aiming to summarize and provide comprehensive ideas for developing the flexible piezoresistive array, this review is presented. The content of this review ranges from piezoresistive materials and microstructures, array design and fabrication, to applications and outlooks. Section 2 focuses on piezoresistive materials and microstructure design. After introducing the performance metrics of pressure sensors, design strategies for performance improvement are discussed in detail. Section 3 focuses on piezoresistive array design and fabrication. According to the resistance readout strategy, i.e., whether active components are integrated with the material, piezoresistive sensor arrays are divided into passive and active arrays. Different structure designs and measuring methods that are capable of providing spatial pressure information are reviewed. There are two main structural forms: one is to arrange multiple independent measurement unit materials and use each piece of material as a measurement point; the other is to make arrayed electrodes on a whole piece of sensitive material and make the electrode intersection point as a measuring point. In addition to the above two forms, other methods, such as visible output and electrical impedance tomography, are also briefly introduced. The crosstalk problem is crucial for the accuracy of spatial measurement. Sources of crosstalk are discussed, including both mechanical and electrical crosstalk. Corresponding suppression approaches include ideas from structure design to readout circuit design. Several fabrication strategies that are promising for mass production are also summarized, consisting of printing, field-assisted processing, and laser-assisted processing. The pressure distribution measured by the sensor device can further provide different information and functions in different application fields. In Section 4, the discussion further focuses on the information that can be extracted from the signals and how they can be used for actual applications, including human-interactive systems, healthcare, and other scenarios. Finally, some outlooks for the development of piezoresistive arrays are provided in Section 5.

## 2. Piezoresistive Materials and Microstructures

### 2.1. Performance Metrics of Pressure Sensors

The evaluation of pressure sensors requires a series of performance metrics, including sensitivity, working range, linearity, hysteresis, response time, relaxation time, and spatial resolution [15]. These indicators are important for understanding the physical properties of the sensor and the sensor selection for specific scene requirements.

The sensitivity of the piezoresistive pressure sensor is defined as the slope of the relative change in resistance or current versus pressure. The quantitative description of sensitivity is as follows, having the unit kPa^−1^ or Pa^−1^.
(1)$$S=\frac{\delta \left(\frac{R-R_{0}}{R_{0}}\right)}{\delta P}\;\mathrm{or}\;S=\frac{\delta \left(\frac{I-I_{0}}{I_{0}}\right)}{\delta P}$$

In general, the sensitivity of many pressure sensors decreases with pressure, and the resistive response saturates beyond the working range; thus, maintaining high sensitivity and linearity over a wide pressure range is an important issue.

Hysteresis influences measurement inaccuracy, manifested as the variation between the loading curve and the unloading curve. The quantitative indicator of hysteresis is defined as
(2)$$\left|\frac{A_{unloading}-A_{loading}}{A_{loading}}\right|\times 100\%\;\mathrm{or}\;\left|\frac{\Delta H_{max}}{y_{FS}}\right|\times 100\%$$
where $\;A_{unloading}$ and $\;A_{loading}$ are the areas under the unloading and loading curve [16,17], $\Delta H_{max}$ is the maximum difference between the loading and unloading curve at the same pressure, and is the full-scale output [15,18,19]. The intrinsic cause of the hysteresis phenomenon includes viscoelasticity of the material, adhesion between the conductive material and substrate, etc. Additionally, most piezoresistive materials based on polymer composite materials have relatively high hysteresis. This is due to the inability of its internal conductive network to fully recover after rearranging under stress [20]. In addition to developing low hysteresis materials, circuitry [21] and algorithms [22] can be applied to compensate for the hysteresis.

Response time and relaxation time (also known as recovery time) are crucial factors for dynamic measurement scenarios. They are used to quantitatively describe the delay of resistance change to pressure change when loading and unloading. Mechanical viscoelasticity is the main cause of the delay [15,23]. 

For the sensor array, the spatial distribution of measurement points can be characterized by spatial resolution, which is represented in the form of N × N [24], the spatial period of the arrangement (center-to-center distance between adjacent pixels) [25], or the number of measuring points per unit area (in dots per inch or pixels per inch) [26]. A detailed discussion about array design and spatial resolution is presented in Section 3.1.2. 

### 2.2. Piezoresistive Materials and Microstructure Design 

Piezoresistive sensors convert pressure into a change in resistance. The resistance can be measured by either sandwiched between the positive and negative electrodes or placed on coplanar electrodes, dominated by either vertical or lateral resistance, respectively. The resistance can be considered as the combination of bulk resistance and contact resistance [27,28]. Bulk resistance is related to material properties, including geometry and internal electrical property. Contact resistance is related to the contact area and interface between the material and the electrodes. To enable sensors with better performance, various materials and sensing structures have been extensively studied.

In terms of piezoresistive materials, besides metal strain gauges and semiconductor devices [7], polymer composite materials have been a focus in the literature [15,29], which shows great prospects in large areas and flexible sensor devices. Polymer composite materials are usually composed of plastic matrix and conductive components. For polymer composites consisting of conductive fillers and insulating substrates, bulk resistance is related to both its geometry and the conductive pathways inside the material, explained by the percolation theory and the tunneling effect [30,31]. The percolation theory analyzes the pressure-induced percolation pathways formed by conductive components, and the tunneling effect refers to the tunneling of charge carriers without contact between particles [7]. Common choices for elastic bases are polymers, e.g., polydimethylsiloxane (PDMS) [32,33], polyurethane (PU) [34], and thermoplastic polyurethane (TPU) [35,36]. Hydrogels [37] are preferred for biocompatible requirements. Conductive components usually include (1) carbon-based materials, e.g., carbon black (CB) [35,38], carbon nanotubes (CNTs) [39,40], graphene [41,42], hybrid carbon fillers [43], etc.; (2) metal materials, e.g., metal nanowires [34,44] and metal particles [26]; (3) others include MXene [45,46], conductive polymers, e.g., polypyrrole (PPy) [16,47], polyaniline (PANI) [48], poly(3,4-ethylene dioxythiophene) (PEDOT) [49], etc. For the electrode material connecting each measurement point of the array, metal materials are usually selected, e.g., Au-based serpentine connections [50] and spay-coated Ag nanowire electrode strips [26]. Adjusting the ratio and concentration of substrates and conductive fillers will lead to different sensor characteristics. Shi et al. reported a pressure sensor with extremely low loading (<1.5 wt.%) of urchin-like hollow carbon spheres in PDMS fabricated by the spin-coating method. The sensor relied on the Fowler–Nordheim tunneling effect, which enabled a large tunneling distance and resulted in an ultrahigh sensitivity of 260.3 kPa^−1^ at 1 Pa. The minimal amount of filler material also endowed the sensor with desirable properties such as high transparency, high elasticity, biocompatibility, and ease of fabrication. Furthermore, the hollow structure contributed to the resistance to temperature variations [51]. By adjusting the ratio of ink components (which consists of conductive carbon nanotubes, insulating silica nanoparticles, and silicone elastomer polymer), Tang et al. proposed a soft and porous composite pressure sensor fabricated by 3D printing technology which can be tuned between negative and positive piezoresistive effect. At a lower CNTs content, a positive piezoresistive pressure sensor with a sensitivity of 0.096 kPa^−1^ across a 0~175 kPa pressure range can be produced with good linearity [52].

In addition to the selection and proportion of materials, the performance of sensors can be further improved by designing three-dimensional microstructures on the surface. The microstructures enhance the material’s compressibility and stress concentration in terms of mechanical properties and facilitate changing the contact area of the conductive interface, thereby improving the sensor’s pressure-sensing performance. In terms of specific surface morphology, both regular and irregular microstructures can be applied. Regular surface structures include microdome [53], micropyramid [54], micropillar arrays [55], conical frustum-like surface structures [56], etc., improving the sensitivity by increasing the contact area under pressure. These microstructures can be fabricated by casting or coating onto templates. The templates can be silica molds prepared by photolithography [57], laser engraved mold [58], etc. Additional surface morphology can also be added to the 3D structure for further improvement. For example, Yao et al. developed a piezoresistive sensor with cracked metallic film coated on the micropyramidal elastomer, which exhibited a sensitivity of more than 10^7^ Ω kPa^−1^ and a low hysteresis of 2.99 ± 1.37% over 0~20 kPa. To create the cracked morphology, they deposited a thin Pt film on PDMS with a micropyramid surface structure and compressed the material. The key to creating regular and annular is by using a soft low-tack adhesive during compression [59]. According to Li et al., the sharp microstructure can be combined with a short electrode channel length to enhance the sensitivity of the piezoresistive pressure sensor. Based on this, they fabricated a sensor with a sharp micropyramid structure and short-channel coplanar Au electrodes (with a channel length of 300 μm and channel width of 1 mm), achieving a sensitivity of 1907.2 kPa^−1^ in the 0~100 Pa range and 461.5 kPa^−1^ in the 100~1000 Pa range, as well as a detection limit of 0.075 Pa and a fast response time of 50 μs [53]. To further improve the sensor performance, irregular surface structures attract interest from researchers, including wrinkles [60], plant-inspired structures using natural biomaterial templates (e.g., leaves [46]), and human skin-inspired structures using randomly distributed spinosum templates (e.g., abrasive paper [42]). Additionally, Zhao et al. reported a self-formed microstructure for a piezoresistive film with a surface roughness of about 8~10 μm, which is much smaller than the controllable microstructures (typically 15~100 μm). The piezoresistive film (PRF) was synthesized by mixing multi-walled carbon nanotubes (MWCNTs) with thermoplastic polyurethane (TPU) elastomer at low temperatures [36]. These irregular surface structures generally have higher performance but are not as good as regular structures in terms of controllability, uniformity, and mass production capacity. Further, a multilayer structure can be designed by stacking two layers of microstructure face-to-face, resulting in an interlocked structure. Examples of multilayer structures include both regularly arrayed microstructures [8] and irregular microstructures stacked face-to-face [42], corresponding to the categories of surface morphologies mentioned above. 

The internal porous structure is another way to improve sensitivity and working range, owing to its lower modulus and higher compressibility. Preparation strategies include freeze-drying [35], salt or sugar template [61], etc. The pores close when applied with pressure, and the conductive surfaces on the pores contact each other to form conductive paths. Conductive components can be additionally coated to enhance performance. For example, Park et al. fabricated low hysteresis porous piezoresistive material with conductive MWCNTs particles coated at the inner surface of the PDMS pores. They explored both the mechanical and piezoresistive hysteresis of the material with values less than 21.7% and 6.8%, respectively [17]. The pores can have uniform arrangement and size, which enhances the sensor-to-sensor consistency compared to structures with random sizes. Oh et al. presented a piezoresistive sensor with uniform porosity, achieving a coefficient of variation of 2.43%. The microfluidic emulsion droplet self-assembly technique was used to fabricate the porous PDMS elastomer, which exhibited uniformly sized pores arranged in a highly ordered, close-packed manner. They also addressed the problem of bonding strength between elastic substrates and conductive materials by chemically grafting conductive polymer (PPy) on the surface of porous elastomer (PDMS) to establish stronger covalent bonds than physical adhesion, achieving a low hysteresis of 2% [16]. Hierarchical pores can also be applied. Inspired by bamboo, Dai et al. developed a hierarchical pore structure in conductive carbon nanofibers (CNFs)/PDMS foam materials to address the conflict between sensor sensitivity and mechanical reliability in porous structures [62]. The hierarchical pore structures, which consist of large-scale pores of several hundred micrometers, hollow structures of several micrometers, and micro/nanoscale irregular pores on the hollow skeleton, respond to tiny pressures by forming additional conductive paths and exhibit high sensitivity of ~0.6 kPa^−1^ at 0~1 kPa. The synergy of porous structure and surface microstructure has also been explored. The synergistic effects of the surface microstructure and the porous structure can enhance the contact area and the number of conductive pathways under applied pressure. For example, Li et al. fabricated a piezoresistive material with high porosity and elliptical surface microstructure by using a mixture of PDMS and MWCNTs, stacking two layers to form an interlock structure. The material exhibited a sensitivity of 10.805 kPa^−1^ in the 1~1000 Pa range and 2.015 kPa^−1^ in the 1 k~100 kPa range [63]. 

Figure 1 provides a summary of the material and structure of flexible piezoresistive pressure sensors.

## 3. Piezoresistive Array Design and Fabrication

### 3.1. Design

#### 3.1.1. Passive and Active Arrays

At the analog front end, the pressure-modulated resistance array is sequentially scanned and converted to a voltage signal by readout circuitry for further analog-digital conversion. According to the readout method, matrix sensor arrays can be classified into passive and active-matrix arrays [64]. In passive arrays, electrodes are laid directly on the piezoresistive material, while in active arrays, active components (e.g., transistors) are tightly integrated with each pixel element. Active arrays have advantages in signal transduction and integration, but passive arrays are easier to fabricate.

For passive-matrix array construction, electrodes are laid directly on the piezoresistive material. The positive and negative ends can be individually led out for each pixel. For larger array density, a row-column structure is preferable due to the reduced number of wires. Resistors in the same row and column share the same wire. Row lines and column lines are selected sequentially to complete a scan of the whole pressure distribution map (Figure 2A). Signal reading and conditioning are carried out in the external circuit design. A voltage divider, negative-feedback amplifier structure, and Wheatstone bridge are three principles for resistor-voltage conversion. Operational amplifiers can be applied to decrease the output impedance, thus reducing the loading effect [6]. Note that for the row-column mode in the passive matrix, electrical crosstalk may exist inside the array, leading to inaccurate measurement of the resistance. Solutions to this issue will be discussed later in Section 3.1.3. However, they usually involve more operational amplifiers in the circuit design, increasing the hardware volume when higher pixel density is applied. Figure 2B,C presents examples of the passive matrix construction and scanning electronics of commercial products (Tekscan Inc., Norwood, Boston, MA, USA). For the active-matrix array construction, each pixel is integrated with a transistor during fabrication. The pressure-sensitive resistor is connected to a transistor in series. Thin film transistors (TFTs) are common choices for flexible sensors with active matrixes (Figure 2D) [5]. A similar row-column selection method is applied. As shown in Figure 2E [65], the rows are connected to the gates, the columns are connected to the drains, and the sources are connected to the ground resistors. When a pixel transistor is turned on and selected, current flows through the resistor to the ground, generating an output voltage. The electrical crosstalk in the active arrays is reduced compared to the passive arrays. However, the parameters of the transistor, e.g., the on–off state current, affect the allowed range of the sensor resistance. 

#### 3.1.2. Structure Designs and Measuring Mechanisms

Spatial resolution is the primary performance metric for evaluating a pressure distribution detection array. There are several structures to measure spatial pressure distribution, and the definition of spatial resolution varies according to the array structure. One direct way is to arrange discrete material elements and corresponding electrodes on substrates. The resolution is commonly reported as the number/density of pixel materials, informs of 3 × 3 [48,52,66], 4 × 4 [8,46,63,67,68], 5 × 5 [62], 6 × 6 [45], and 8 × 8 [25]. An example is shown in Figure 3A [62]. However, the uniformity of pixel property and the gaps between pixels need to be considered [16,69]. For some scenarios, such as decease diagnostic, it is preferable to place individual pressure sensing elements at several feature points (e.g., arches and joints) and interpret the information through further algorithms without a neat row-column array [6]. Another way is to cover the electrodes of the array with a continuous piece of piezoresistive materials to form the array [26,36], which is capable of achieving much higher spatial resolution. Their resolution is often reported as the density of readout points, that is, the density of electrode intersections in passive arrays and the density of transistors in active arrays. For this configuration, the spatial resolution is limited by both the sensing material and the electrodes. For example, Shi et al. developed a pressure sensor based on Fowler–Nordheim tunneling effect by spin-coating urchin-like hollow carbon spheres in polydimethylsiloxane (PDMS) with concentration far below the percolation threshold. The material forms a vertical conduction path under pressure while being horizontally insulated, thereby reducing transverse interference. Theoretically, the sensing density can be 2,718,557 per cm^2^; however, limited by electrode size, a 64 × 64 passive matrix was fabricated in a 32 × 32 mm area by photolithography, reaching a sensing density of 400 cm^−2^. The thin-film pressure sensor exhibits a high sensitivity of 260.3 kPa^−1^, high transparency, and reduced temperature interference (Figure 3B) [51].

Improving resolution while maintaining other good performances is of great importance. As mentioned previously, microstructures commonly involve an increase in sensitivity by introducing variations in the contact area. The size of the microstructure influences pixel density, considering uniformity and flatness. Regular structures (e.g., micropyramids) typically fall in the range of 15~100 μm. Irregular structures based on special molds, such as plants, exhibit better performance but are not suitable for large-scale production; thus, the array demonstrations in these essays are usually individual elements arranged in individual forms. Recently, Zhao et al. developed a large-scale piezoresistive sensor with a high spatial resolution of 0.9 mm (28.2 ppi) by applying a kind of MWCNTs/TPU material with a self-formed surface structure on the scale of 8~10 μm. The microstructure was flat enough for the 0.9 mm pixel size and had a high sensitivity of ~385 kPa. The material was integrated into a 64 × 64 active matrix using CNT TFTs, covering a 4-inch area (Figure 3C) [36]. Researchers also focus on other features, such as air permeability which is required for wearable applications. Pei et al. fabricated a high-resolution array with a porous structure by 3D printing. Silicone ink was extruded by gas to form a porous structure, and graphene was attached to the surface as the piezoresistive material. The sensor had a linear response in the range of 0~12 kPa and a sensitivity of 4 Pa^−1^. The pitch distance between each sensing unit is 1 mm, equivalent to a resolution of 100 cm^−2^ (Figure 3D) [70]. Besides higher resolution, multiple resolution, an important sensing characteristic of human skin, was also researched. Mimicking real tactile sensing of human fingers, Kim et al. developed a multiple-resolution piezoresistive sensor by arranging pitch-varying electrodes on a single piece of piezoresistive material. Using aligned Ni/PDMS material, the resolution of their sensor can be adjusted up to 100 dpi with a pitch distance from 0.25 mm to 1 mm (Figure 3E) [26]. In addition to focusing on sensor structure, appropriate processing algorithms will greatly improve the sensing limit under the same hardware conditions [71].

Besides fabricating pixel structures by either assembling discrete material elements or arranging electrodes, there are also pixel-less methods that do not employ patterned pixels on physical structures. One is a light-omitting sensor device with visible output. Lee et al. proposed a two-layer structure consisting of a pressure-sensing film lying on an electroluminescent film capable of displaying high-resolution images corresponding to the pressure distribution. The top film was coated with a kind of cathode and cellulose/nanowire nanohybrid network for pressure sensing, which controls the current flowing through the quantum-dot light-emitting diode on the bottom film, thus influencing the image display. The spatial resolution was over 1000 dpi (evaluated by loading the micro-bumps array). The sensor had a sensitivity of over 5000 kPa^−1^ and a response time of less than 1 ms. The displayed image was captured by high-resolution cameras in real-time, and the pressure data can be resolved by pre-calibrated image data, which avoided local data acquisition and processing electronics (Figure 3F) [72]. Another is the electrical impedance tomography (EIT) method. It is a technique that can reconstruct the conductivity of the internal area only through electrodes at the boundary. Current is injected into the conductive film, and the voltage is analyzed to reconstruct the impedance distribution [73,74]. An example is shown in Figure 3G [75]. The quality of the result depends largely on the reconstruction algorithm. The EIT method simplifies the internal wiring of the sensing material, thus making it much easier for manufacturing. However, it suffers from poor spatial resolution and low temporal frequency [76].

**Figure 3 polymers-15-02699-f003:**
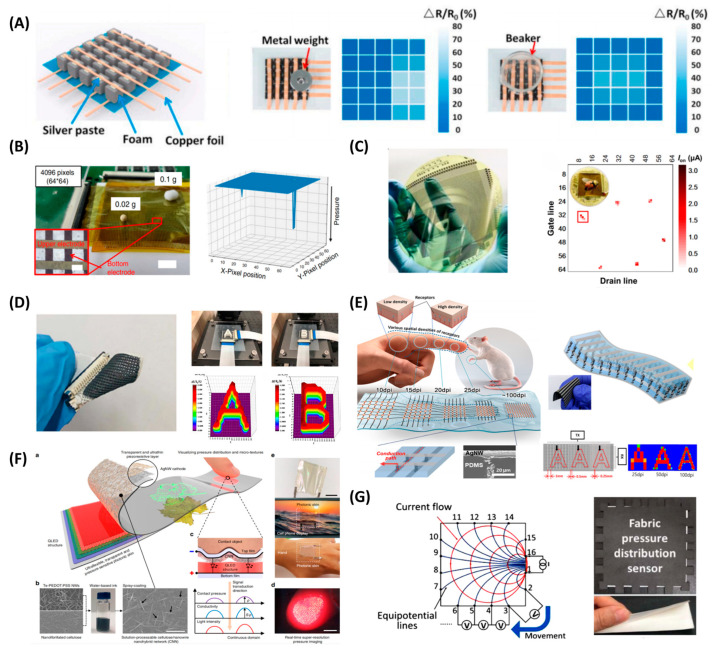
Structure designs and measuring mechanisms: (**A**) Example of arranging discrete material elements with 1.0 × 1.0 cm^2^ pixel size. Reproduced with permission from [62]. Copyright 2021, Elsevier. (**B**) Quantum effect-based flexible and transparent pressure sensors with a sensing density of 400 cm^−2^. Reproduced with permission from [51]. Copyright 2020, Springer Nature. (**C**) A large-scale piezoresistive sensor with a high spatial resolution of 0.9 mm (28.2 ppi) by applying MWCNTs/TPU material with a self-formed surface structure. Reproduced with permission from [36]. Copyright 2022, American Chemical Society. (**D**) Air permeable pressure sensor array with a resolution of 100 cm^−2^. Reproduced with permission from [70]. Copyright 2021, Elsevier. (**E**) A multiple-resolution piezoresistive sensor having a pitch distance from 0.25 mm to 1 mm (up to 100 dpi). Reproduced with permission from [26]. Copyright 2022, American Chemical Society. (**F**) Light-omitting sensor device with visible output. Reproduced with permission from [72]. Copyright 2020, Springer Nature. (**G**) Electrical impedance tomography (EIT) method. Reproduced with permission from [76]. Copyright 2020, IEEE.

In summary, a generic strategy is to attach sensing elements separately to substrates, providing flexible choices for sensor displacement. For pixel-based structures, most previous works stuck at millimeter range resolution with a resolution of no higher than 100 dpi in the literature. This is probably limited by the distance between the conductive path or microstructure size of the material and the fabrication ability of flexible electronics. Pixel-less methods ease the requirement of locally arrayed electronics but involve additional devices or complex algorithm analysis. 

#### 3.1.3. Crosstalk and Suppression

It is worth noting that the spatial resolution of the pressure detection array is not only determined by the array structure. All arrays face the challenge of crosstalk. The coupling between the array elements may introduce errors in the pressure measurement of each point, known as crosstalk interference. Thus, for the evaluation of the sensor array, it is necessary to determine maximum resolution under the condition that interference among pixels falls below a certain level [26] and to reduce crosstalk to achieve higher measurement accuracy.

Since both the mechanical and electrical response of each sensing pixel may be influenced by other pixels, the sources of crosstalk issues can be categorized into mechanical crosstalk and electrical crosstalk [25]. Mechanical crosstalk is derived from deformation coupling. A force applied to one pixel would generate deformation of the adjacent pixels, resulting in the resistance change in the unloaded pixels. This is a noteworthy problem for the common sensor structure that covers a whole piece of piezoresistive material on arrayed electrodes [77]. Arranging discrete sensing elements is a direct solution [33], but the elastic supporting layer may still be a cause. To evaluate this issue, Li et al. studied three interconnection methods: serpentine, straight line, and unpatterned piece. They found that serpentine interconnects could best suppress crosstalk between adjacent pixels (Figure 4A) [25]. Electrical crosstalk was caused by unintended conductive pathways or current leakage, resulting in an inaccurate readout of the resistance. The electrical crosstalk paths have two sources. One is the lateral conductive path inside the material at different measuring points, and the other is the crosstalk loop of the external resistance reading circuit of the material. A detailed introduction and countermeasures are presented in the following paragraphs.

For the first condition, an isolated piezoresistive pixel structure can be applied [50,78]. Coplanar electrodes, such as comb structures, can also suppress crosstalk [25,33]. Apart from attaching individual sensing elements, researchers also find some solutions by designing special geometry structures. Park et al. introduced a grooved structure between parallel electrodes through the molding process, which increased the leakage resistance between adjacent electrodes, thus effectively attenuating the crosstalk interference (Figure 4B) [17]. Kyubin Bae et al. further developed a mesh-structured anisotropic material to reduce lateral conduction sensor array using the dip-coating method. The CNT/PDMS composite was uniformly patterned in the holes and isolated by the mesh layer, eliminating the electrical crosstalk effect and mechanically connecting the sensing elements (Figure 4C) [79]. From the perspective of developing new material, Kim et al. presented an anisotropic material to reduce lateral conduction. They prepared Ni/PDMS mixture and applied a magnetic field to align the nickel particles in the field direction, forming filamentous conduction paths. They discovered that after alignment, the composite had less densely connected conduction paths in the lateral direction. There was virtually no crosstalk at pitches beyond 0.25 mm, achieving tunable resolutions up to 100 dpi (Figure 4D) [26].

For the second condition, array indexing is commonly applied for reading out large sensor arrays, requiring only m + n electrodes. However, since each sensing element is not insulated, this row-column selection method would introduce unintended loops. For example, as shown in Figure 4E [80], in order to read the resistance of R22, the driving voltage was applied at the second row, and the generated voltage was read out at the terminal of the second column. This can lead to unintended currents flowing through the path, shown by the dashed line, which causes inaccuracy of the result. One way to solve the problem is to insert control devices, such as transistors, into the array. Tanaka et al. [81] introduced transistors to choose the sensitive elements and realized a 128 × 128 resistive bolometer array with a scanning rate of 30 Hz. Diode can also be used to control the elements. By placing each element in series with the diode, the parasitic parallel current was suppressed by the single guiding pass of the diode. A high readout rate can be achieved; however, the measurement accuracy is deeply affected by the parameter of the inserting devices, which are susceptible to multiple environmental factors. In addition, the complex construction adds to the complexity and cost of manufacturing. Another way is to build an external circuit to gate the sensing element and short the unselected elements to attenuate the influence of the unintended loops. To eliminate the crosstalk, different measurement methods were applied, such as the voltage feedback methods (VFM) and the zero potential methods (ZPM). The application of VFM was introduced by Tise [82], who realized a 16 × 16 piezoresistive sensor array with a scanning rate of 10 Hz. As shown in Figure 4F [83], the output voltage is feedback to column wires and row wires through an operational amplifier. The voltage drop of nonscanned elements was very small. The idea of ZPM is illustrated in Figure 4G [84]; the output of each column is connected in series with an operational amplifier whose purpose is to set the voltage V_ref_ at the tracks of all columns. Since the voltage of all rows that do not contribute to the output is also set to V_ref_, all of the unintended loops are short-circuited. Both VFM and ZPM have the advantages of high accuracy, fast readout rate, and low interconnection complexity; however, they lead to an increase in the circuit volume. The algorithmic correction method determines the sensor resistance by establishing and solving the resistance matrix equations of the sensor array, avoiding dedicated hardware design. Shu et al. realized a 10 × 10 tactile resistive sensor array with a scanning rate of 30 Hz and a measurement accuracy of 0.61 ± 0.41% [85]. Algorithm correction can greatly reduce the complexity and production difficulty of hardware circuits, but in large arrays, the computational complexity rises sharply, and the accuracy is significantly affected by calibration and analog-digital conversion accuracy. 

In summary, crosstalk derives from both mechanical and electrical aspects. The suppression can focus on novel materials [26], geometry structure design [25,79], external circuit design [60,62], or backend algorithmic compensation [85].

**Figure 4 polymers-15-02699-f004:**
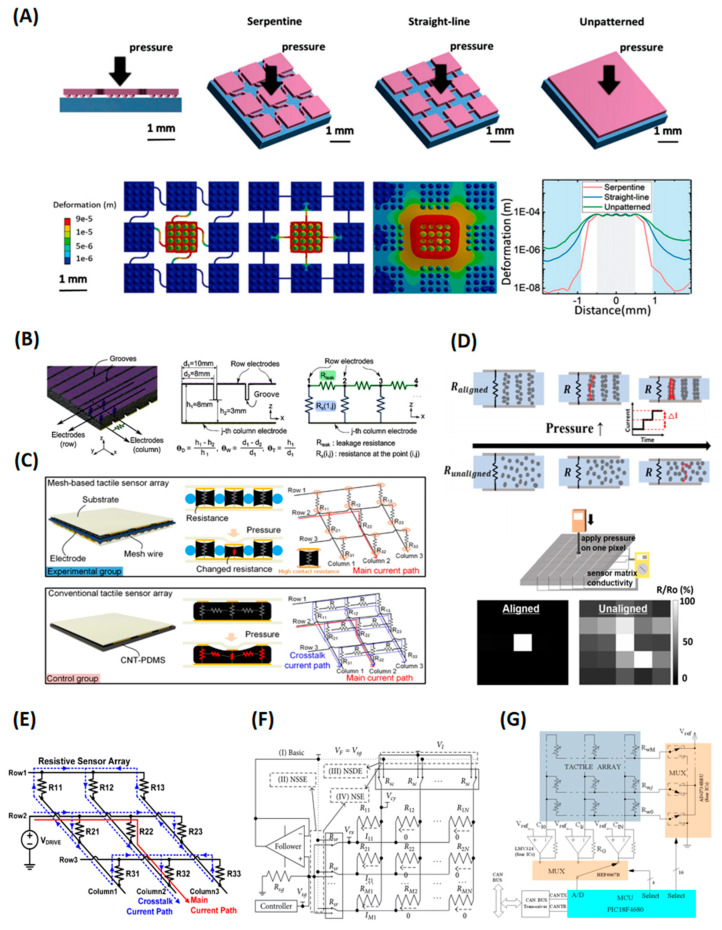
Crosstalk and suppression: (**A**) Mechanical crosstalk. Reproduced with permission from [25]. Copyright 2022, John Wiley and Sons. (**B**) Grooved structure for electrical crosstalk reduction. Reproduced with permission from [17]. Copyright 2019, Elsevier. (**C**) Mesh structure for electrical crosstalk reduction. Reproduced with permission from [79]. Copyright 2021, American Chemical Society. (**D**) Anisotropic material for electrical crosstalk reduction. Reproduced with permission from [26]. Copyright 2022, American Chemical Society. (**E**) Mechanisms of electrical crosstalk caused by unintended circuit loops; (**F**) schematic of the voltage feedback methods; (**G**) schematic of the zero potential methods.

### 3.2. Fabrication Strategies

#### 3.2.1. Printing

Roll-to-roll printing, screen printing, and inkjet printing are common printing methods for piezoresistive sensors. Roll-to-roll printing and screen printing are only suitable for 2D patterns, while inkjet printing is more versatile and maskless. 

In inkjet printing, proper ink is prepared and directly printed on the receiver, forming a 2D or 3D structure. Inkjet 3D printing technology is also called direct ink writing, which is a layer-by-layer printing technique based on the pressure-driven deposition of ink through a nozzle [86,87]. Compared with other additive manufacturing technology, ink writing offers more diversity in the choice of materials. There is an ongoing appearance of novel materials for direct ink writing, e.g., polymers, metal, composites, etc. A review is presented in [87]. Inkjet printing is an attractive fabrication technique for flexible and thin film devices at a low cost. Both the sensing material and electronics can be fabricated by inkjet printing. For piezoresistive material, carbon-based nanomaterials such as CNTs are often added to the polymer to form composite ink with good electrical properties. For example, Tang et al. reported a soft and porous piezoresistive composite by 3D printing (Figure 5A). The ink was formed by dispersing CNTs and silica nanoparticles in a silicone elastomer solution [52]. Using 2D inkjet printing, Lee et al. developed an all-paper pressure sensor [88], as in Figure 5B. The carbon nanotubes were diluted in water, and printed on both sides of the stacked mulberry paper as sensing elements, followed by the drying process. Silver nanoparticles were printed as electrodes. The sensor exhibits a sensitivity exceeding 1 kPa^−1^ over a wide pressure range of 0.05~900 kPa.

Electronic components can also be fabricated by inkjet printing. Besides printed electrodes [88], printed thin film transistor (TFT) technology has been making progress in materials and methodology [89]. TFTs are promising for the integration of active pressure sensing arrays with low crosstalk and high flexibility. Using inkjet printing, Baek et al. fabricated 10 × 10 TFT arrays with various resolutions (up to 10.6 pixels per inch), achieving high uniformity (less than 10% relative standard deviation) and 100% yield. The electrodes are inkjet-printed using Ag nanoparticle ink, and perylene was deposited as the gate dielectric layer. Hydrophobic Teflon was printed as a bank line, and organic semiconductor ink was printed to fill the bank area. The TFT array was further integrated with piezoresistive sheets in a one-transistor-one-resistor structure, forming a wearable pressure sensor matrix [5].

#### 3.2.2. Field-Assisted Processing

For solution-based fabrication, simple dip-coating or casting cannot guarantee a controlled positioning and alignment of the nanostructure and fillers inside the material. Electric fields and magnetic fields can be involved in the fabrication process for better manipulation.

There are multiple manufacturing technologies based on the electric field, including electrospinning, electroplating (aka., electrodeposition), electrospraying, etc. Among them, electrospinning is a mature and fast method for nanofiber production and alignment, which can be applied to produce flexible piezoresistive materials, as shown in Figure 6A,B. During electrospinning, high voltage was applied to the syringe, forming a Taylor cone. The charged threads were drawn from the syringe towards the grounded collector by the electric force, thus producing the nanofiber [90]. Conventional electrospinning is mostly used to generate 2D thin film mesh structure. For example, Li et al. fabricated a flexible piezoresistive sensor based on thermoplastic-urethane (TPU) decorated with carboxyl MWCNTs (c-MWCNTs), forming an electrospun fibrous network (~0.15 mm thickness) with good conductivity. The material has a sensitivity of 2 kPa^−1^, in the 10 kPa pressure range, and a hysteresis below 6% [39]. Electrospinning for 3D porous structures has also attracted attention. Han et al. applied electrospinning and thermal carbonization to fabricate a 3D porous carbon nanofiber network as a piezoresistive sensor in a simple and low-cost way [91] (Figure 6A). They prepared a precursor solution composed of polyacrylonitrile, dimethylformamide, and AlCl_3_. After electrospinning, the floc was preoxidized and then carbonized into the final porous carbon nanofiber networks. The material had a sensitivity of 1.41 kPa^−1^, stable resilience, and high compressibility. Kweon et al. fabricated PVDF-HFP/PEDOT nanofibers by 3D electrospinning and vapor deposition polymerization [49] (Figure 6B). The material had a sensitivity of 13.5 kPa^−1^ and a thickness of ~1 mm. It was further integrated into a 16 × 10 array in an 8 cm × 6 cm area for spatiotemporal pressure mapping.

The magnetic field can also assist in the alignment of fillers in composite solutions, as shown in Figure 6C,D. Composite piezoresistive materials with good properties can be achieved after magnetic field treatment. The advantage of the magnetic field is that it is cheap and easy to generate, e.g., just by permanent magnets. However, limited materials have a response to it, e.g., nickel, iron, carbon nanotubes, etc. For example, Kim et al. applied a magnetic field to align the nickel particles filled in the PDMS matrix prior to the curing process, forming an anisotropic percolation path that reduced the crosstalk. A piezoresistive film sensor with a spatial resolution of up to 100 dpi was fabricated with the material [26] (Figure 6C). Wang et al. applied a magnetic field to align the direction of magnetic rGO@nickel nanowires in the EcoFlex matrix during curing, resulting in a low percolation threshold of 0.27% and transmittance of 71.8%. The microdome microstructure was further involved by a hot embossing template. The sensor exhibits a sensitivity of 1302.1 kPa^−1^ [92] (Figure 6D).

**Figure 6 polymers-15-02699-f006:**
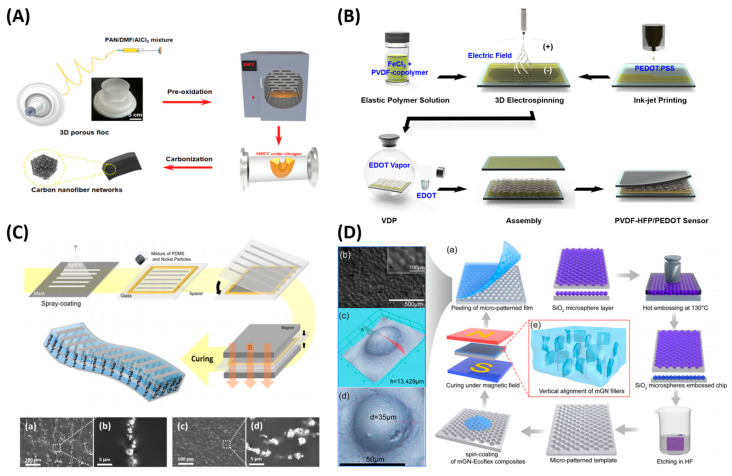
(**A**) An example of electric field-assisted processing from [91]. (**B**) An example of electric field-assisted processing from [49]. (**C**) An example of magnetic field-assisted processing from [26]. Reproduced with permission from [26]. Copyright 2022, American Chemical Society. (**D**) An Example of magnetic field-assisted processing from [92]. Reproduced with permission from [92]. Copyright 2019, American Chemical Society.

#### 3.2.3. Laser-Assisted Processing

Laser is widely applied in today’s manufacturing. With different material properties and light parameters, the laser can induce different effects in the material, e.g., heating, ablation, carbonization, polymerization, etc. 

Carbonaceous piezoresistive material can be generated from a polymeric precursor by laser irradiation. A typical example is laser-induced graphene (LIG), aka., laser-scribed graphene (LSG) [93,94]. One of the most popular precursors is polyimide, and the laser can be, e.g., infrared (IR) laser [95] or ultraviolet (UV) laser [25]. The property of the induced graphene material depends on the laser parameters, and its morphology is characterized by a porous pattern. The geometry of the sensor can be shaped by laser manufacturing. Laser cutting and Laser engraving are mature fabrication strategies. Laser cutting uses a high-power laser to cut through the material to form various shapes [25,50]. Laser engraving partially removes the material, fabricating controllable surface microstructure as templates [58]. Conventional silicon-based lithography and etching can also be applied to produce molds with precise morphologies; however, the devices are more expensive [57,96]. The following is representative of recent work using lasers for material fabrication and geometry. Li et al. used a UV laser to fabricate a serpentine piezoresistive array in two steps (Figure 7). In the first step, polyimide film was partially converted to isolated 3D porous graphene pixels by low-power UV laser (2.1~3.8 W). In the second step, higher laser power (6.8 W) was applied to cut the substrate layer by ablation to form the serpentine connection. Finally, an 8 × 8 array was created with a pixel size of 0.4 mm and a pixel distance of 0.3 mm [25].

## 4. Applications

### 4.1. Human-Interactive System

Human–computer interaction (HCI) is central to realizing efficient collaboration between humans and the digital world, and the data acquisition part of HCI inevitably relies heavily on pressure input [97]. Traditional input devices are less conformable and portable, e.g., keyboard, gamepad, and mouse. With the increasing requirements for acquisition accuracy, portability, and seamless interaction, flexible piezoresistive sensor arrays have more advantages in monitoring human movement, providing more possibilities for a new generation of human–computer interaction. In this section, we will introduce touchpads and tactile gloves, covering both control input devices and interactive systems. 

Electronic skin patches and multipixel touchpads based on flexible sensor arrays are becoming a mainstream trend in HCI research [98]. They have the advantages of high sensitivity, compact size, good stability, and a relatively simple manufacturing process. Here, we introduce tactile keyboards and smart touchpads. To build a tactile keyboard, Pyo et al. introduced a tactile sensor comprising stacked carbon nanotubes and Ni-fabrics (Figure 8A) [99]. Flexible electronics based on multifunctional fabrics have attracted great interest for their light weight, flexibility, and easy fabrication [100]. The hierarchical structure of the fabric increases the contact area and distributes the stress to each layer of the fabric, improving sensitivity and linearity. The sensor presents a sensitivity of 26.13 kPa^−1^ over a pressure range of 0.2~982 kPa. The sensors were used to build a full-size all-fabric keyboard which included 29 tactile sensor cells corresponding to all 26 letter keys, a comma, a dot, and a space bar (Figure 8A). The sensor array consisted of 14 electrical lines constructed by the row-column selection wiring method. Smart touchpads can also be applied for more complex input recognition, such as handwritten digits and characters. Using a tactile sensor with 64 × 64 pixels, Zhao et al. collected 3099 images of handwritten digits, realizing a classification accuracy of 98.8% for testing. More complex Chinese characters were also collected. They selected 9 Chinese characters and collected a dataset of 900 images (100 for each). Different percentages of the handwritten data set were added to the computer-generated training set (12,150 images), and the recognition accuracy reached 97.3% (Figure 8B) [36].

Smart tactile gloves have become one of the most concerning branches of human–computer interaction. With human hands performing most of the tasks in everyday work, projecting hand movements and using them to control robots or virtual objects has become key to achieving immersive experiences in work, play, and learning. Currently, the relatively mature technology in smart gloves is the use of inertial measurement units (IMU) to measure hand movement. This kind of glove offers excellent accuracy and repeatability of angular motion tracking of fingers owing to the advanced MEMS technology [101]. However, purely IMU-based gloves lack applied force monitoring, especially when looking for information about touch. Therefore, piezoresistive material-based sensors have become an important supplement for smart glove sensing. By assembling flexible piezoresistive arrays, pressure distribution patterns during object grasping and manipulation can be studied, revealing how humans interact with the environment. Sinha et al. attached piezoresistive arrays on fingertips to form a smart glove that had 45 sensing points (3 × 3 array on each of the five fingers) to collect pressure information. Deep learning was used to identify sharp and blunt objects and the direction of the pressure with a classification accuracy of 95.9% and 97.8%, respectively [77] (Figure 9A). Charalambides et al. developed a glove capable of detecting both normal and shear forces, as shown in (Figure 9B) [102]. Each node of the sensor consisted of pillar and pad structures using CNT and PDMS. Different loading forces caused different contact statuses between the pillar and pad, which caused different contact resistances. Developing wearable smart gloves and ensuring their functionality require sensors to be robust to common forms of human hand movement, including large deformations and exposure to complex environments. Gao et al. reported a microfluidic tactile diaphragm pressure sensor based on embedded Galinstan microchannels (Figure 9C) [103]. Four sets of pressure sensors, including two tangential and two radial sensors, were connected end-to-end to form an equivalent Wheatstone bridge, providing highly sensitive output and temperature self-compensation. The proposed tactile sensor had a sensitivity of 0.0835 kPa^−1^ and a limit of detection of 100 Pa with sub-50 Pa resolution. Smart glove based on this sensor was capable of providing dynamic responses toward a variety of hand motions such as holding, gripping, grasping, squeezing, lifting, moving, or touching objects. Aiming to learn the full-hand tactile information of human grasping, Sundaram et al. fabricated a tactile glove with 548 sensing points covering the full hand. The glove was fabricated by attaching piezoresistive film (commercial conductive polymer, 3M Velostat) on the knitted glove, forming a passive matrix by arranging orthogonal electrodes. Using this tactile glove, they studied spatial and temporal tactile information when manipulating 26 objects and recorded a large-scale dataset with 135,000 frames. By training a deep convolutional neural network, typical tactile patterns of grasping motions were analyzed; object identification and weighing are demonstrated [4] (Figure 9D). 

Further, combining tactile perception with remote feedback can be used to build remote sensing systems. The system usually includes a sensor at the remote end, a communication device for haptic data feedback, and an actuator as stimuli at the user end. For example, in Figure 9E, researchers developed a tactile interface consisting of the tactile glove, a linear-actuator-based tactile display, and two microcontroller units (MCUs) for data processing and wireless Bluetooth communication. The tactile information sensed by user A was measured by the wearable tactile glove, while simultaneously, the actuator-based tactile display provided tactile information to user B [99]. The construction can also be built into a teleoperation system [104,105], which is the user-side control of a robot in a remote scene to perform tasks, such as the manipulation of objects, which has great potential for various applications, e.g., telerobotic surgery and virtual reality. The tactile feedback is crucial for the planning of the movement and controlling of applied force.

Since arrayed sensors increase spatial information, more information about the contact can be explored. For the extraction of information from a large amount of pressure data, the rapid development of machine learning provides an effective way [106]. By utilizing appropriate learning models for specific sensing applications, more comprehensive information can be extracted for sensors that are simply designed, such as motion sequences, touch forces, and slides [107,108]. Based on training, the output patterns of grasping or touching behaviors of different objects, gestures and object recognition can be realized instead of simple motion detection at the primary level [109,110]. This will further build a more intelligent human-interactive system. In addition, machine learning algorithms can be carried out not only by computers after collecting the data but also in artificial-intelligent hardware for near-sensor computing. Memristor has been widely studied for building neuromorphic devices. A prototype implementation of the memristor-based compute-in-memory chip with a tactile sensor array is demonstrated in [36]. 

**Figure 9 polymers-15-02699-f009:**
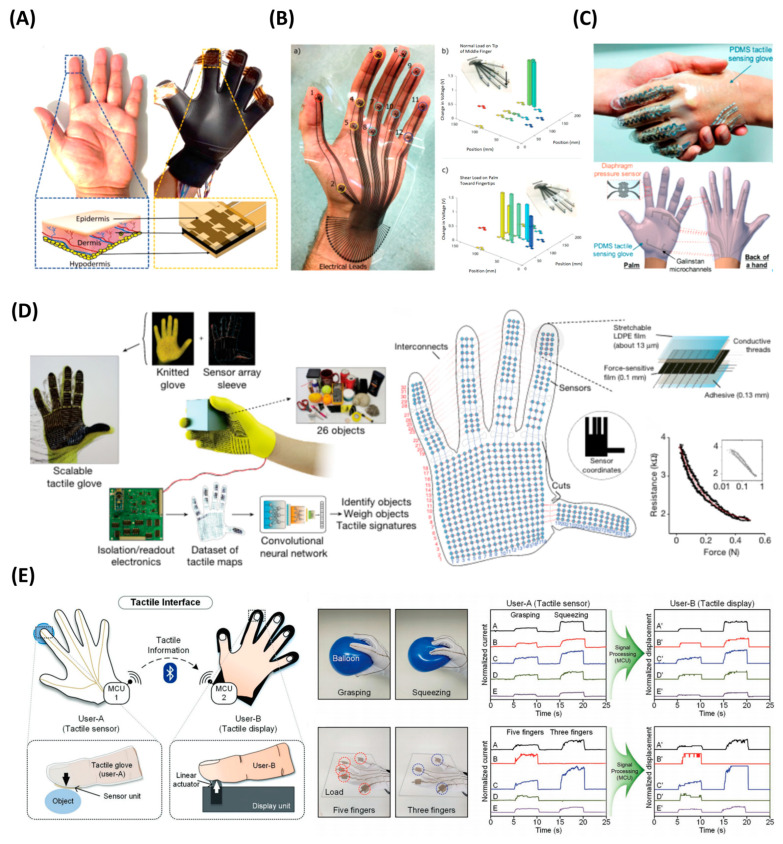
Glove-based human–machine interfaces: (**A**) Tactile glove with piezoresistive arrays on fingertips. Reproduced with permission from [77]. Copyright 2022, John Wiley and Sons. (**B**) Tactile glove capable of detecting both normal and shear forces. Reproduced with permission from [102]. Copyright 2016, John Wiley and Sons. (**C**) A microfluidic tactile diaphragm pressure sensor-based tactile glove capable of providing dynamic responses toward a variety of hand motions. Reproduced with permission from [103]. Copyright 2017, John Wiley and Sons. (**D**) Tactile glove for learning the full-hand tactile information of human grasping. Reproduced with permission from [4]. Copyright 2019, Springer Nature. (**E**) A tactile interface consists of the tactile glove, a linear-actuator-based tactile display, and two microcontroller units (MCUs) for data processing and wireless Bluetooth communication. Reproduced with permission from [99]. Copyright 2019, John Wiley and Sons.

### 4.2. Healthcare

Pressure sensors can be applied in hospitals, where large datasets are easy to collect, thus, are promising for the application of machine learning algorithms to help automatic diagnosis. Plantar and pulse measurements are two main applications of the pressure sensor array in the medical field. Additionally, artificial skin based on sensor arrays is also often used in prosthetics to reconstruct the skin’s sensing response to the vital characteristic of external stimuli.

The measurement of plantar pressure distribution has a wide range of applications, including disease diagnosis, rehabilitation, athletic analysis, etc. For plantar pressure distribution measurement, typical requirements of the performance include (1) pressure range: over 1000 kPa in waking analysis, 1900 kPa in daily activities, and 3000 kPa in some extreme situations; (2) pressure resolution less than 10 kPa is widely accepted for insole, sometimes below 1 kPa (e.g., gait abnormality in diabetes); (3) bandwidth for dynamic pressure: 100 Hz for walking and 200 Hz for running and jumping; (4) a general element size of 5 mm is recommended by [111]. Considering the arrangement of sensor elements, there are two strategies. One is to measure the pressure distribution across the whole plantar area, and the other is placing sensing elements at several feature points. Several mature commercial products have been developed with the former strategy, e.g., the F-scan system by the Tekscan company (Figure 10A). Although these products provide sufficient information and are versatile, they have high costs, and the high density may be redundant. Related to this issue, many research works focus on developing new high-performance materials and arranging them at feature positions to demonstrate their usability for actual plantar measurement. Gait patterns are further extracted from the pressure distribution and analyzed by algorithms (e.g., machine learning) for diagnosis. For example, Xiao et al. fabricated polyvinylchloride/carbon black microgrid films for wearable applications. The sensor has a sensitivity of 4.71 kPa^−1^ in the working range of up to 15 kPa and a response time of 5 ms. An insole pressure sensing array with 24 units was realized. Three kinds of feet (normal foot, high arch foot, and flatfoot) and three walking gait phases (heel striking, standing, and pushing-off) can be identified [112] (Figure 10B). Monitoring plantar pressure is also helpful for patients with neural diseases that cause peripheral neuropathy or abnormal gait, e.g., diabetes [113], Parkinson’s disease [114,115], etc. 

Wrist pulse signals can detect multiple health information, e.g., heart rate, blood pressure, vascular function, etc. Wearable pressure sensors for wrist pulse detection are widely studied. Since the pulse is weak and short impulse signal, new materials developed for pulse monitoring target high sensitivity and fast response. Noting that higher sensitivity may amplify both the target signal and the noise, it is also necessary to take signal-noise ratio (SNR) into evaluation when optimizing the sensor sensitivity (e.g., adjusting the filler proportion). Considering the stability and reliability of pulse monitoring, sensors with moderate sensitivity instead of ultrahigh sensitivity, are preferable [116]. Previous sensors are mostly individual sensor measurements, e.g., across the artery or specific positions according to traditional Chinese medicine [19]. Although exhibiting high performance, the optimal sensor position needs to be carefully determined due to the thin diameter of the radial artery. Compared with a single sensor, arrayed pressure sensors can better locate the arterial line and obtain a higher-quality signal for the assessment. Huang et al. proposed a linearly arranged 8 × 1 piezoresistive array for blood pulse monitoring above a radial artery. For each element, the length-width ratio is above eight and aligned along the artery direction for higher sensitivity. The sensor was fabricated by combining two layers of CNT/PDMS composites with interlock microdomes, exhibiting a sensitivity of −6.08 kPa^−1^ (Figure 10C) [117]. A 2D matrix has also been proposed. Baek et al. generated a spatial-temporal map of the pulse by integrating customizable thin film transistor arrays with piezoresistive film. It had a pressure sensitivity of 16.8 kPa^−1^ in the range of less than 1 kPa, 1.25 kPa^−1^ in the range of more than 2 kPa, and low power consumption of 10 nW. To monitor the cardiac condition, they observed the signals of each pixel in a 5 × 5 matrix to locate the artery line and extracted the augmentation index, which was an indicator of arterial stiffness (Figure 10D) [5].

Artificial skin constructed from piezoresistive sensor arrays can mimic many of the mechanical properties of human skin and realize a skin-like sensation, which provides several key advantages in helping amputees adapt to the use of prostheses and avoid injury [118].For example, providing sensory feedback from the prosthesis can make users more inclined to feel that the prosthesis is part of the body, promoting a sense of ownership [119]; stimulating the residual sensory pathways with sensory information can reduce phantom limb pain of up to 80% amputees [120]; providing tactile feedback can also make the operation of the prosthesis more natural and easier for patients [121,122]. Ferreira et al. developed a polymer-based piezoresistive sensor array that can assess the pressure distribution at the Interface of prosthetic alveolar in real-time. The sensor arrays enabled practitioners to better evaluate the matching degree of the socket and improve the performances of prosthetic alignment and training with a comfortable feeling for the patients (Figure 10E) [123]. Tactile feedback is important for the functional improvement of the prosthesis, Kim et al. [124] developed an artificial afferent nerve based on multiple piezoresistive sensors, realizing the connection to biological efferent nerves and demonstrating the information flow to actuate the tibial extensor muscle in the leg (Figure 10F). Osborn et al. [125] proposed a multi-layered, sensorized synthetic skin for prosthesis applications, which can provide a full map of tactile information feedback for the limb.

**Figure 10 polymers-15-02699-f010:**
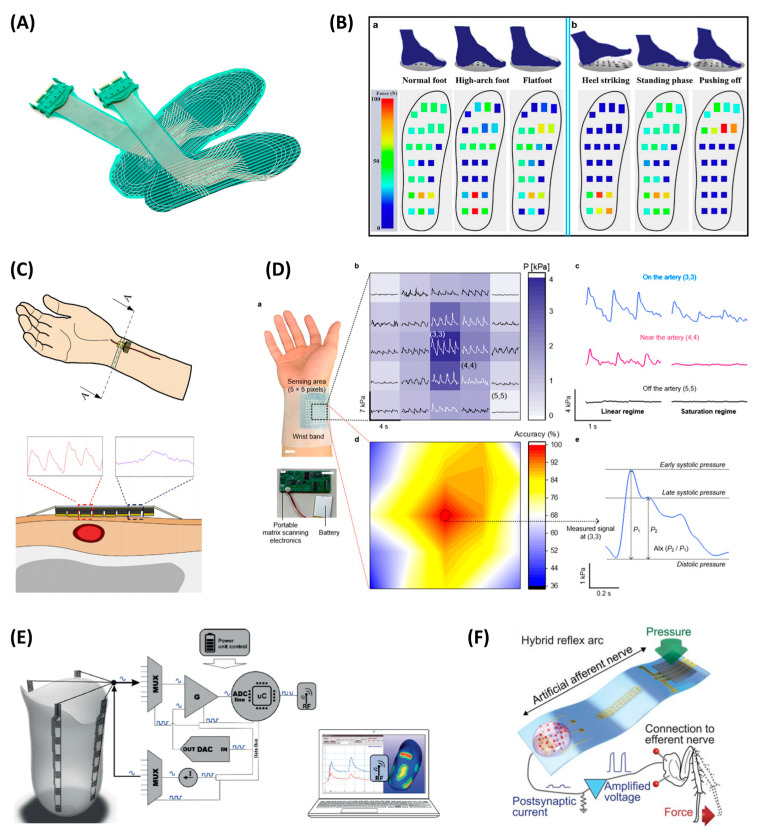
Healthcare applications: (**A**) F-scan system for plantar pressure measurement by the Tekscan company. (**B**) Plantar application of piezoresistive array to identify three kinds of feet (normal foot, high arch foot, and flatfoot) and three walking gait phases (heel striking, standing, and pushing-off). Reproduced with permission from [112]. Copyright 2019, IOP Publishing. (**C**) A linearly arranged 8 × 1 piezoresistive array for blood pulse monitoring above a radial artery. Reproduced with permission from [117]. Copyright 2018, Elsevier. (**D**) A 5 × 5 matrix for generating a spatial-temporal map of the pulse by integrating customizable thin film transistor arrays with piezoresistive film. Reproduced with permission from [5]. Copyright 2022, American Chemical Society. (**E**) Piezoresistive polymer-based sensors for assessing pressure distribution at the prosthetic interface. Reproduced with permission from [123]. Copyright 2017, IEEE. (**F**) An artificial afferent nerve based on multiple piezoresistive sensors. Reproduced with permission from [124]. Copyright 2018, Science.

### 4.3. Others

The measurement of pressure distribution also has applications in underwater conditions, for example, in the marine environment. Marine biologging has been studied. By attaching sensors to marine animals, their behaviors and marine environment can be explored [126]. Bulky and rigid sensors may influence animal behavior; thus, a conformable and lightweight sensor is required. Moreover, the arrayed sensor can provide more detailed water conditions. Flexible pressure sensor arrays are promising for marine biologging. Nassar et al. proposed a fully conformable multi-sensing “marine skin” with a resistive type temperature sensor array and capacitive type pressure sensor array, which is 55 mm × 55 mm × 0.3 mm in size and weight less than 2.4 g (Figure 11A) [127]. Similar to the thin-film and flexible properties and better tolerance to humidity, piezoresistive sensor arrays could be promising for marine biologging. Besides biology and ecology, flow monitoring can also be applied to marine vehicles. In the water environment, the interaction between the hull and the water will affect the travel and maneuvering of the ship. Inspired by fish, real-time monitoring and feedback of the pressure distribution along the lateral line can assist marine vehicles in traveling better. Dusek et al. developed a four-by-one piezoresistive array in an 80 cm × 20 cm area for the pressure monitoring of unmanned marine vehicles with low-cost and waterproof (Figure 11B). They used carbon black-PDMS foam as the sensing material, with filler concentration near the percolation threshold. For underwater characterization, the sensor was attached to an automatic control platform which generated oscillation to create dynamic pressure. The signal was amplified and filtered by a 112.88 Hz low pass filter. Experiments showed that the sensor had a dynamic range of 50~500 Pa and a pressure resolution of 5 Pa [128].

## 5. Conclusions and Outlooks

In conclusion, this review comprehensively summarizes the considerations for developing flexible piezoresistive array devices, from the material and microstructure for performance enhancement to the array design and fabrication strategies, providing recent advances. The significance and application methods of flexible piezoresistive array devices to practical scenarios are also reviewed, providing ideas for more integrated sensing systems. Some typical and promising examples of piezoresistive array configurations in recent years and their performances are presented in Table 1. Array structures are classified into two categories. One is arranging discrete piezoresistive material elements in space, with each material element as a sensing pixel. The other is applying a continuous piece of piezoresistive material, with the intersection of electrodes as a sensing pixel. The material, microstructure, and corresponding performances of these typical researches are listed.

Future efforts can focus on the following aspects. First, improving the performance of flexible piezoresistive material is a constant issue, especially the unignorable problem of hysteresis that influences the accuracy of dynamic measurement; however, the amount of research work is limited. Future works should focus more on theories about hysteresis and develop adaptable materials with lower hysteresis. Second, polymer composites that are commonly used in flexible piezoresistive arrays exhibit poor selectivity under environmental interference, especially temperature [130,131]. The intrinsic reason is that the thermal expansion coefficient of the polymer matrix is usually higher than that of the filler, which leads to a mismatch in the thermal expansion between the two materials, causing internal variation when temperature changes. This is a vital problem for practical applications where pressure sensors may be required all year round or in extreme conditions. Third, although many high-performance new materials have only been made into single-point devices in research, they have the potential to be developed into high-performance array sensors through structural design methods mentioned in Section 3.1.2, which can further endow them with more versatility. Fourth, although the property of flexibility allows the sensors to be attached on a curved surface, pre-stress and pre-strain caused by the attachment surface may lead to the drift of resistance and degradation of electronic components (e.g., transistors). Fifth, the improvement of spatial resolution is constrained by crosstalk issues. While increasing the array density, attentions need to be paid to whether more significant crosstalk is introduced between adjacent pixels. Sixth, although materials with novel microstructure designs exhibit higher performance, uniformity and controllability are essential considerations for large-area and mass production. In terms of fabrication, efforts are still needed from laboratory production to industrial manufacturing, where the emphasis is on cost and time efficiency. Finally, in terms of application, most researches focus on robotics and wearable healthcare scenarios; however, pressure sensor arrays can assist in multiple scenarios where single-point pressure sensors are required. Despite the lack of precision and maturity, the relative spatial distribution can help extract new and complementary information.

## Figures and Tables

**Figure 1 polymers-15-02699-f001:**
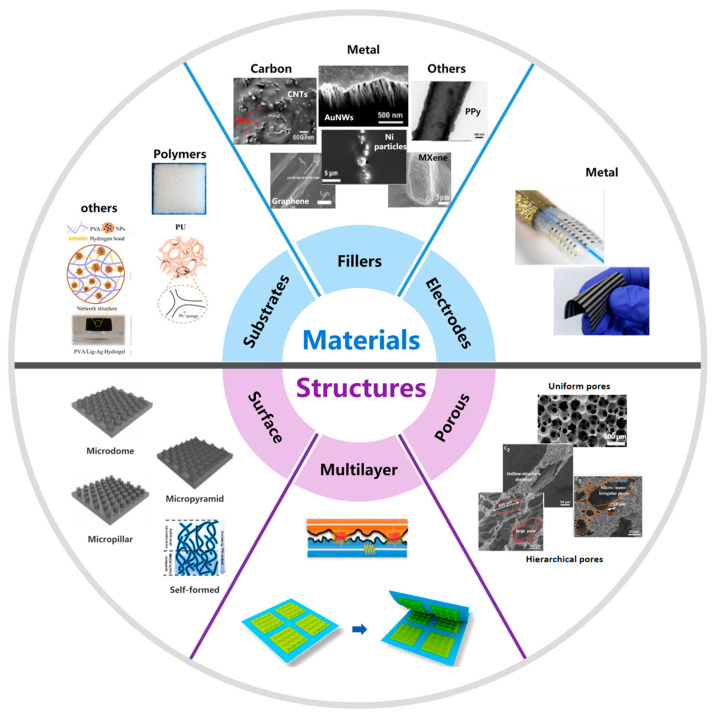
Piezoresistive materials and microstructure design. In terms of substrates, common choices of materials are polymers [34] (Copyright 2019, American Chemical Society). Others include hydrogel [37] (Copyright 2021, Elsevier). In terms of fillers, examples of carbon-based materials include “CNTs” [40] (Copyright 2019, American Chemical Society); “Graphene” [42] (Copyright 2018, American Chemical Society); examples of metal-based materials include “Au nanowires (AuNWs)” [44] (Copyright 2019, American Chemical Society) and “Ni Particles” [26] (Copyright 2022, American Chemical Society); other materials include “PPy” [47] (Copyright 2022, Elsevier) and “MXene” [46] (Copyright 2022, Elsevier). In terms of electrodes, metal materials are usually selected, for example, Au-based serpentine connections [50] (Copyright 2020, Springer Nature) and spay-coated Ag nanowire electrode strips [26] (Copyright 2022, American Chemical Society). Microstructures are designed to improve performance, including surface structure, multilayer structure, and porous structure. Examples of surface structures include arrayed microdome, micropillar, micropyramid [55] (Open Access), and self-formed irregular structures [36] (Copyright 2022, American Chemical Society). Examples of multilayer structures include regularly arrayed interlocked microstructures [8] (Open Access) and stacked irregular structures [42] (Copyright 2018, American Chemical Society). Examples of porous structures include uniform pores [16] (Copyright 2019, John Wiley and Sons) and hierarchical pores [62] (Copyright 2021, Elsevier).

**Figure 2 polymers-15-02699-f002:**
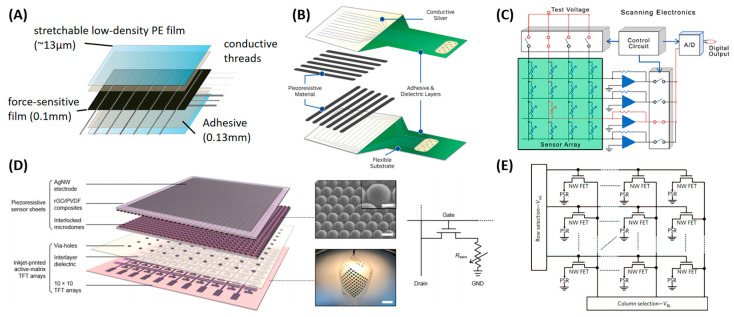
Passive and active matrix construction: (**A**) Example of the passive matrix. Reproduced with permission from [4]. Copyright 2019, Springer Nature. (**B**) Example of passive array construction in a commercial product (Tekscan). (**C**) Example of passive array scanning electronics in a commercial product (Tekscan). (**D**) Example of TFT-based active-matrix construction. Reproduced with permission from [5]. Copyright 2022, American Chemical Society. (**E**) Example of readout approach of the active matrix. Reproduced with permission from [65]. Copyright 2010, Springer Nature.

**Figure 5 polymers-15-02699-f005:**
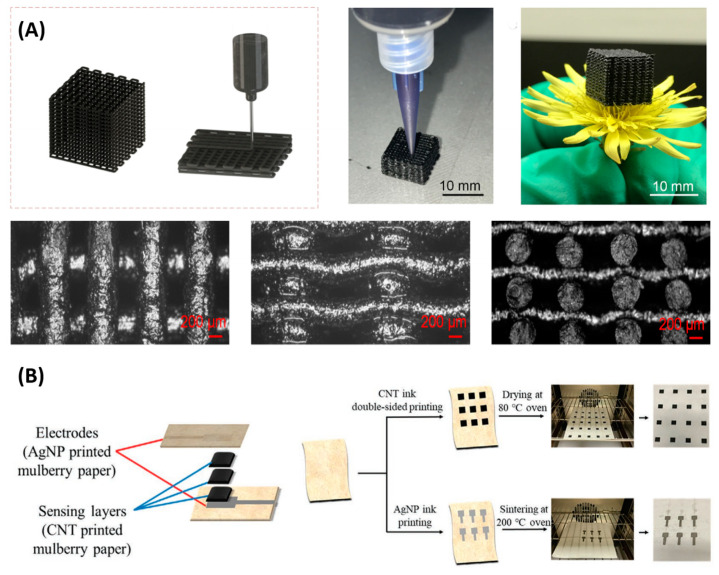
(**A**) Example of 3D printing and the microscope images of the structure . Reproduced with permission from [52]. Copyright 2022, American Chemical Society. (**B**) Example of 2D printing. Reproduced with permission from [88]. Copyright 2021, John Wiley and Sons.

**Figure 7 polymers-15-02699-f007:**
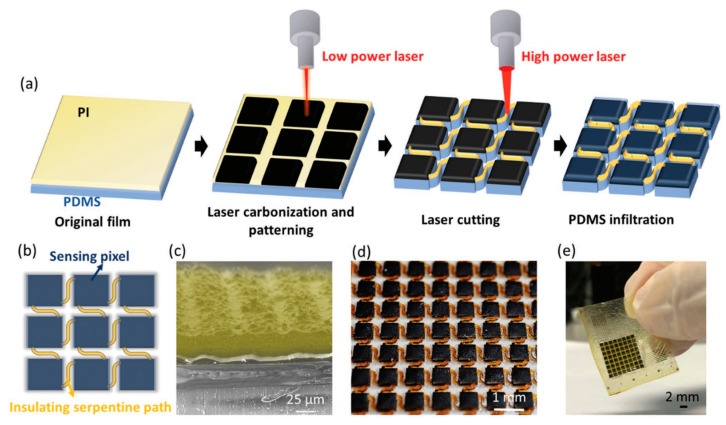
An example of laser processing for piezoresistive sensor fabrication. (**a**) Fabrication process. (**b**) Top view of the active layer. (**c**) Scanning electron microscopy (SEM) image of the active layer. (**d**) Picture of the material array. (**e**) Picture of an encapsulated pressure sensor array. Reproduced with permission from [25]. Copyright 2022, John Wiley and Sons.

**Figure 8 polymers-15-02699-f008:**
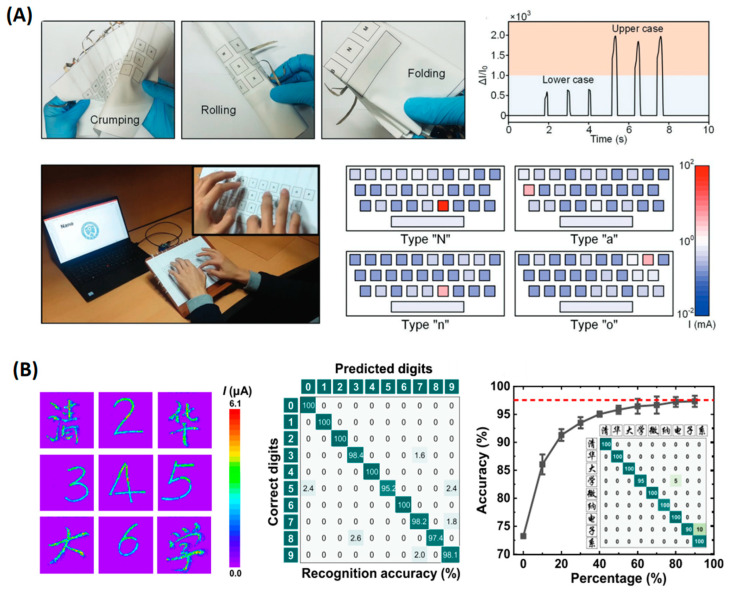
(**A**) Tactile keyboard. Reproduced with permission from [99]. Copyright 2019, John Wiley and Sons. (**B**) Smart touchpads for handwritten digits and character recognition. Reproduced with permission from [36]. Copyright 2022, American Chemical Society.

**Figure 11 polymers-15-02699-f011:**
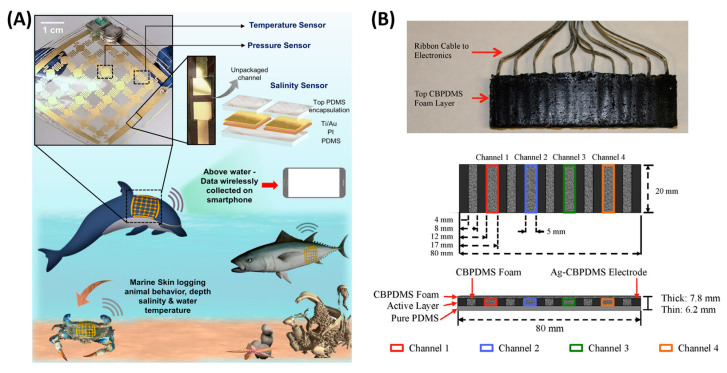
Other applications: (**A**) marine biologging [128]; (**B**) flow monitoring. Reproduced with permission from [129]. Copyright 2016, Elsevier.

**Table 1 polymers-15-02699-t001:** A summary of typical and promising examples of piezoresistive arrays.

Array Structure	Spatial Resolution	Material	Microstructure	Sensitivity and Working Range	Response Time/Relaxation Time (ms)	Ref.
Discrete material elements	3 × 3 array, pixel size: 3 × 3 mm^2^, center-to-center distance: 4.5 mm	graphene ink on patterned PDMS	lotus leaf, two layers stacked face to face	1.2 kPa^−1^ (pressure range 0~25 kPa)	/	[66]
	4 × 4 array in 6 cm × 6 cm area	carbon nanostructure on patterned PDMS	anisotropic wavy microstructures, two layers stacked face to face	1.214 kPa^−1^ (pressure range: 0~100 Pa) 0.301 kPa^−1^ (pressure range: 0.1~1 kPa)	266/766	[129]
	5 × 5 array, pixel size: 1.0 × 1.0 cm^2^	carbon nanofibers/PDMS	hierarchical pore structures	0.60 kPa^−1^ (pressure range: 0~1 kPa) 0.08 kPa^−1^ (pressure range: 1~6 kPa) 0.01 kPa^−1^ (pressure range: 6~20 kPa)	30/25	[62]
	6 × 6 array	MXene on patterned PDMS	sandpaper, two layers stacked face to face	2.6 kPa^−1^ (pressure range: 0~30 kPa)	40/40	[45]
	8 × 8 array, pixel size: 0.4 mm, center-to-center distance: 0.7 mm	laser-induced graphene foam	/	1.37 kPa^−1^ (pressure range: 80 kPa)	20/~30	[25]
	10 × 10 array in 7 cm × 7 cm area	MXene@ P(VDF-TrFE)	/	817.4 kPa^−1^ (pressure range: 0.072~0.74 kPa) 2213.68 kPa^−1^(pressure range: 0.74~3.083 kPa)	16/16	[24]
Continuous piece	6.6, 8.1, 10.6 ppi	rGO/PVDF	interlocked microdomes	16.8 kPa^−1^ (pressure range < 1 kPa) 1.25 kPa^−1^ (pressure range > 2 kPa)	<200	[5]
	64 × 64, 28.2 ppi	MWCNTs/TPU	self-formed surface structure	10 kPa^−1^ (pressure range 15~1400 kPa)	5/3	[36]
	up to 100 dpi	aligned Ni/PDMS	/	0.72 kPa^−1^ at 357 kPa working range of up to 373 kPa	~12/~20	[26]
	400 cm^−2^	hollow carbon spheres/PDMS	/	260.3 kPa^−1^ at 1 Pa >1 kPa^−1^ (pressure range: 1~800 Pa) >0.1 kPa^−1^ (pressure range: 800~10,000 Pa)	60/30	[51]

## Data Availability

Not applicable.

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
