# Peer review of "Recent Advances in Flexible Piezoresistive Arrays: Materials, Design, and Applications"

_polymers, 2023, doi:10.3390/polym15122699_

Round 1

Reviewer 1 Report

The manuscript reports on the Recent Advances in Flexible Piezoresistive Arrays: Materials, Design, and Applications. The manuscript is communicated in a lucid way, and it could be useful for researchers working in the area of resistance-based sensors. However, the manuscript requires minor changes and additions before its acceptance.

1.     The performance metrics of the piezo-resistive sensor are well-known (Section 2.1). Therefore, I recommend authors make it brief or eliminate the section if necessary.

2.     Kindly, comment on the benefits and limitations of piezoresistive sensors over piezo-capacitive sensors and their practicality.

3.     Piezoresistive arrays show poor selectivity due to their ability to respond to humidity, gases, temperature, etc. Therefore, authors are requested to discuss briefly the same, which may help the readers to get the overall perspective of the piezoresistive sensors. 

The language quality is good and readable. However, authors are requested to go through the typographical errors meticulously.

Reviewer 2 Report

Recent Advances in Flexible Piezoresistive Arrays: Materials, Design, and Applications 

Authors point of view the content of this review ranges from piezoresistive materials and microstructures, array design and fabrication, to applications  and outlooks. Also, focuses on piezoresistive materials and microstructure design.

English is well written

Well organized, with adequate figures and a comprehensive table.

Figures are with atmost clarity.

Table is also complete as required for the researchers.

Finally, some outlooks for the development of piezoresistive arrays are provided.

This review comprehensively summarizes the considerations for developing flexible piezoresistive array devices, from the material and microstructure for performance enhancement to the array design and fabrication strategies, providing recent advances. The significance and application methods of flexible piezoresistive array devices to practical scenarios are also reviewed, providing ideas for more integrated sensing systems.

Accept it in its present form

Author Response

Thank you for your comment, we will further improve our manuscript and strive for publication.

Reviewer 3 Report

The manuscript "Recent Advances in Flexible Piezoresistive Arrays: Materials, Design, and Applications" as review is very well written, There only few points to address

1. please check that all abbreviations are defined as example page 7 line 263 TFTs

2. The part of healthcare applications need additional parts such as artificial skin where such sensors are very interesting to apply. If there is not enough research done such might fit in the outlook

3. It also would be beneficial add a Table after application or before outlook of such sensors which are most promising and what is there range as well sensitivity as pressure sensor.
